# 11β-Hydroxysteroid Dehydrogenase Type 1 within Osteoclasts Mediates the Bone Protective Properties of Therapeutic Corticosteroids in Chronic Inflammation

**DOI:** 10.3390/ijms23137334

**Published:** 2022-06-30

**Authors:** Chloe G Fenton, Ana Crastin, Claire S Martin, Saicharan Suresh, Isabella Montagna, Bismah Hussain, Amy J Naylor, Simon W Jones, Morten S Hansen, Caroline M Gorvin, Maria Price, Andrew Filer, Mark S Cooper, Gareth G Lavery, Karim Raza, Rowan S Hardy

**Affiliations:** 1Institute for Metabolism and Systems Research, University of Birmingham, Birmingham B15 2TT, UK; chloe.fenton@uni-wuerzburg.de (C.G.F.); csm765@student.bham.ac.uk (C.S.M.); isabella.montagna@yahoo.co.uk (I.M.); c.gorvin@bham.ac.uk (C.M.G.); mxp797@student.bham.ac.uk (M.P.); gareth.lavery@ntu.ac.uk (G.G.L.); 2Research into Inflammatory Arthritis Centre Versus Arthritis, Institute of Inflammation and Ageing, University of Birmingham, Birmingham B15 2TT, UK; a.naylor@bham.ac.uk (A.J.N.); a.filer@bham.ac.uk (A.F.); k.raza@bham.ac.uk (K.R.); 3Institute of Clinical Science, University of Birmingham, Birmingham B15 2TT, UK; a.crastin@bham.ac.uk (A.C.); sxs1916@student.bham.ac.uk (S.S.); bxh975@student.bham.ac.uk (B.H.); 4MRC Arthritis Research UK Centre for Musculoskeletal Ageing Research, University of Birmingham, Edgbaston Campus, Birmingham B15 2TT, UK; s.w.jones@bham.ac.uk; 5Molecular Endocrinology Laboratory (KMEB), Department of Endocrinology, Odense University Hospital, DK-5000 Odense, Denmark; morten.steen.hansen@rsyd.dk; 6Centre for Membrane Proteins and Receptors (COMPARE), Universities of Birmingham and Nottingham, Birmingham B15 2TT, UK; 7ANZAC Research Institute, The University of Sydney, Sydney, NSW 2006, Australia; mark.cooper@sydney.edu.au; 8Department of Rheumatology, Sandwell and West Birmingham NHS Trust, Birmingham B15 2TT, UK

**Keywords:** 11β-hydroxysteroid dehydrogenase type 1, inflammatory bone loss, corticosteroids, polyarthritis, osteoclasts, rheumatoid arthritis

## Abstract

Therapeutic glucocorticoids (GCs) are powerful anti-inflammatory tools in the management of chronic inflammatory diseases such as rheumatoid arthritis (RA). However, their actions on bone in this context are complex. The enzyme 11β-hydroxysteroid dehydrogenase type 1 (11β-HSD1) is a mediator of the anti-inflammatory actions of therapeutic glucocorticoids (GCs) in vivo. In this study we delineate the role of 11β-HSD1 in the effects of GC on bone during inflammatory polyarthritis. Its function was assessed in bone biopsies from patients with RA and osteoarthritis, and in primary osteoblasts and osteoclasts. Bone metabolism was assessed in the TNF-tg model of polyarthritis treated with oral GC (corticosterone), in animals with global (TNF-tg^11βKO^), mesenchymal (including osteoblast) (TNF-tg^11βflx/tw2cre^) and myeloid (including osteoclast) (TNF-tg^11βflx/LysMcre^) deletion. Bone parameters were assessed by micro-CT, static histomorphometry and serum metabolism markers. We observed a marked increase in 11β-HSD1 activity in bone in RA relative to osteoarthritis bone, whilst the pro-inflammatory cytokine TNFα upregulated 11β-HSD1 within osteoblasts and osteoclasts. In osteoclasts, 11β-HSD1 mediated the suppression of bone resorption by GCs. Whilst corticosterone prevented the inflammatory loss of trabecular bone in TNF-tg animals, counterparts with global deletion of 11β-HSD1 were resistant to these protective actions, characterised by increased osteoclastic bone resorption. Targeted deletion of 11β-HSD1 within osteoclasts and myeloid derived cells partially reproduced the GC resistant phenotype. These data reveal the critical role of 11β-HSD1 within bone and osteoclasts in mediating the suppression of inflammatory bone loss in response to therapeutic GCs in chronic inflammatory disease.

## 1. Introduction

Due to their potent anti-inflammatory and immune modulatory properties, therapeutic glucocorticoids (GCs) such as hydrocortisone and prednisolone are widely utilised in the management of inflammatory diseases, including ulcerative colitis, Crohn’s and rheumatoid arthritis (RA) [1,2,3]. However, their application is severely limited by off target side effects such as muscle wasting and GC induced osteoporosis (GIOP) that increase morbidity and mortality [4,5,6]. The mechanisms that underpin GIOP in the absence of inflammation are well defined and are characterised by a reduction in osteoblast numbers function. However, the effects of GCs on bone in the context of systemic inflammation (the scenario in which therapeutic GCs are typically used) are more complex [7,8,9]. In this situation, their anti-anabolic actions in osteoblasts are offset by a suppression of bone resorption by osteoclasts [8]. These bone protective effects are attributed to both a direct action on osteoclast, as well as through the suppression of local pro-inflammatory mediators such as IL-1β, IL-6, IL-8, IL-12, IL-17 and TNFα, which otherwise increase osteoclast numbers and function (either directly or through actions on the bone regulatory receptor activator of nuclear factor kappa-Β ligand (RANKL) and osteoprotegerin (OPG) pathways) [8,10,11,12].

The GC activating enzyme, 11β-hydroxysteroid dehydrogenase type 1 (11β-HSD1), activates endogenous and synthetic GC precursors (such as cortisone and prednisone in humans, 11-dehydrocorticosterone (11-DHC) in mice) to their active counterparts (hydrocortisone, prednisolone and corticosterone), amplifying their actions within target cells [13,14]. Murine models with transgenic deletion of 11β-HSD1 have revealed that this enzyme is a critical mediator of both the beneficial anti-inflammatory, and deleterious off target side effects of therapeutic corticosteroids at sites of inflammation and in tissues such as muscle and bone [9,15,16]. However, whilst 11β-HSD1 is expressed in both osteoblasts and osteoclasts in vivo, its contributions within these cells to the beneficial anti-inflammatory and off target deleterious actions of therapeutic corticosteroids in bone during chronic inflammation remains poorly defined [17,18]. This is of, particular, relevance given the well reported inflammatory upregulation of 11β-HSD1 within mesenchymal and myeloid derived cell populations in vitro and within sites of inflammation [19,20,21,22]. In the absence of inflammation, the global deletion of 11β-HSD1 prevents GIOP following oral GC administration, by preventing the GC induced reduction in osteoblast numbers and function [9]. These data point to the importance of 11β-HSD1 within osteoblasts in regulating bone metabolism in conditions of GC excess, but provide limited insight into its role in a chronic inflammatory setting. Still less still is known of the regulation and functional role of 11β-HSD1 within osteoblasts and osteoclasts in mediating the bone protective actions of therapeutic corticosteroids during systemic inflammation.

In this study we characterise the inflammatory regulation and function of 11β-HSD1 within human bone and in primary bone cell cultures, and using the TNF-tg murine model of chronic polyarthritis and systemic inflammation, we delineate the cellular contribution of bone cells to the bone protective actions of therapeutic corticosteroids in vivo. Using animals with global and cell targeted transgenic deletion of 11β-HSD1, we reveal a critical role of 11β-HSD1 within osteoclasts in mediating the systemic bone protective actions of therapeutic GCs during chronic inflammation.

## 2. Methods

### 2.1. Human Tibia Bone Biopsies

Adult patients with hip osteoarthritis (OA) or rheumatoid arthritis (RA) consented to collection of trabecular bone biopsies (150–500 mg) during elective joint replacement surgery, following ethical approval (REC reference 14/ES/1044 and NRES 16/SS/0172). Current glucocorticoid therapy was an exclusion criterion. Fresh bone tissue was used immediately for enzymatic activity analysis.

### 2.2. Models of Polyarthritis

Procedures were performed under guidelines by the Animal (Scientific Procedures) Act 1986 in accordance with the project licence (P51102987) and approved by the Birmingham Ethical Review Subcommittee (BERSC). The TNF-tg model of chronic inflammatory polyarthritis, obtained courtesy of Professor George Kollias (BSRC Fleming, Athens, Greece), was maintained on a C57BL/6 background and compared to WT littermates [23]. At day 32 of age, at the first onset of measurable polyarthritis, male TNF-tg mice received drinking water supplemented with either corticosterone (Cort) (100 μg/mL, 0.66% ethanol), or vehicle (0.66% ethanol) for 3 weeks. Mice were scored as previously described [24,25]. At day 53, animals were culled by cervical dislocation following a cardiac bleed under terminal anaesthetic and tissues excised, weighed, and fixed in 4% formalin or snap-frozen in liquid nitrogen for later analyses.

### 2.3. Targeted Deletion of 11β-HSD1

11β-HSD1 KO animals with global 11β-HSD1 deletion were crossed with TNF-tg animals to generate TNF-tg^11βKO^ animals as previously described [26]. Mesenchymal 11β-HSD1 KO animals were generated by crossing flx/flx-HSD11B1 mice with Twist2-cre mice to generate 11βHSD1flx/flx/Twist2cre animals, which were paired with TNF-tg animals to produce TNF-tg^11βHSD1flx/flx/Twist2cre^ (TNF-tg^11βflx/tw2cre^) [27,28,29]. Myeloid targeted 11β-HSD1 KO animals were generated by crossing flx/flx-HSD11B1 mice with LysM-cre mice to generate 11βHSD1flx/flx/LysMcre animals which were paired with TNF-tg animals to produce TNF-tg^11βHSD1flx/flx/LysMcre^ (TNF-tg^11βflx/LysMcre^) [30].

### 2.4. 11β-HSD1 Enzymatic Activity Assay

Confluent cells or fresh ex vivo trabecular bone tissues (up to 50 mg) were incubated in medium containing cortisone (100 nmol/L) (for human samples) or DHC (100 nmol/L) (for rodent samples) along with tracer amounts of tritiated cortisone or tritiated DHC (Perkin Elmer, Beaconsfield, UK). Steroids were extracted in dichloromethane and separated by thin-layer chromatography with ethanol/chloroform (8:92) as the mobile phase. Thin-layer chromatography plates were analysed with a Bioscan imager (Bioscan, Washington, DC, USA), and the fractional conversion of steroids was calculated. The protein concentration was determined with the Bio-Rad Protein assay using the Bradford method (Bio-Rad, Hercules, CA, USA). Experiments were performed in triplicate, and enzymatic activity was reported as pmol product per mg protein or mg of tissue per hour as appropriate.

### 2.5. Primary Human Osteoblast Culture

Following ethical approval (UK National Research Ethics Committee 14/ES/1044), patients with hip osteoarthritis (OA) (age 69 ± 3 years, Kellgren Lawrence grade 3/4; *n* = 4) were recruited prior to elective joint replacement surgery. Trabecular chips of approximately 400–600 mg were excised and placed in PBS prior to culturing. Reagents were obtained from Sigma (Gillingham, UK) unless otherwise stated. Trabecular bone chips from patient samples were cultured in osteoblast growth media (Dulbecco’s Modified Eagle Medium with 4500 mg/L glucose (Sigma, Gillimgham, UK) supplemented with 10% fetal bovine serum (FBS), 1% non-essential amino acids, 1% sodium-pyruvate (100 mM) and 1% penicillin-streptomycin (10,000 units P, 10 mg S per mL); and 0.1% ascorbic acid (50 µg/mL) and β-glycerophosphate (2 mM) to facilitate release of osteoblasts. Osteoblasts were allowed to grow, and once confluent bone chips were removed. Osteoblasts were then differentiated in media containing TNFα (10 ng/mL) and/or cortisol (1000 nmol/L). Treatments were replaced three times per week. Cultures were stained with 0.5% alizarin red S at pH to 4.1–4.3 to confirm differentiation into mature osteoblasts.

### 2.6. Primary Human Osteoclast Culture

Peripheral blood mononuclear cells (PBMCs) from healthy donors, obtained from the National Blood Transfusion Service (approved by Birmingham NHS Trust Ethics Committee), were isolated via RosetteSep™ (Miltenyi) Human Monocyte Enrichment Cocktail for negative selection of monocytes. Monocytes were cultured in selective survival media consisting of αMEM media (Gibco) supplemented with 10% foetal bovine serum (FBS), 1% penicillin-streptomycin (10,000 units P, 10 mg S per mL) and M-CSF (20 ng/mL) (Bio-Techne). Osteoclast differentiation was induced through the addition of RANKL (25 ng/mL) (Bio-Techne) 24 h after monocyte isolation and maintained for 6 days before stimulation with vehicle, TNFα (10 ng/mL) or of cortisone (1000 nmol/L) (or DMSO vehicle) as appropriate. Following stimulation, osteoclast cultures were fixed in 10% neutral buffered formalin. Cells were washed twice in PBS before incubation for 40 min at 37 °C in TRAP staining solution consisting of sodium acetate anhydrous (0.1 M), L+ tartaric acid (0.076 M), glacial acetic acid 3%, fast red violet LB salt (0.5 g/mL) and naphthol AS-MX Phosphate substrate Mix (0.5 mL per 100 mL) at pH 4.7–5.0. Cells were then washed twice with in PBS. Area, perimeter and number of osteoclasts were analysed using Image J. Light microscopy at 10× magnification across 5 × 500 µm^2^ fields of view.

### 2.7. In Vitro Bone Resorption Assay

Following differentiation, osteoclasts cultures were detached by incubating in Accutase™ (BioLegend, San Diego, CA, USA) at 37 °C for 8 min and seeded directly onto bovine dentine chips to evaluate in vitro osteoclast functionality. Osteoclasts were seeded directly onto cortical bone slices (0.4 mm) (BoneSlices.com, Jelling, Denmark) in a 96 well plate at 2000 cells per well. Osteoclasts were incubated on bone chips, in the presence of either TNFα (10 ng/mL), cortisone (1000 nmol/L) or both for 48 h at 37 °C. Bone chips were then washed in PBS and gently scraped to remove osteoclasts using a cotton bud. Bone erosions were visualised following staining with 1% toluidine blue solution for 15 s, followed by washing in 70% Ethanol. Bone chips were individually scored across the surface using light microscopy (20× magnification). Bone erosions were scored within 3 categories (Small Pits < 20 µm = 1, medium trenches < 50 µm = 2 and large trenches > 50 µm = 3). Scores from at least three chips per intervention, across 4 independent donors were assessed.

### 2.8. Analysis of P1NP and CTX-1 by ELISA

Serum was collected from mice by cardiac puncture under terminal anaesthetic. Briefly, whole blood was left at room temperature for 30 min prior to centrifugation for 20 min at 12,000 rpm. Serum was aspirated and stored at −80 °C prior to analysis. Serum-free (non-corticosteroid binding globulin bound) corticosterone levels were measured using a commercially available sandwich ELISA designed to specifically detect active (but not inactive 11-DHC) steroid (cat no: KGE009, R&D systems, Abingdon, UK). Serum was analysed in accordance with the manufacturer’s instructions and data expressed as ng/mL. Serum P1NP was determined using a commercially available sandwich ELISA (cat no: AC-33F1, Immunodiagnosticsystems, Tyne & Wear, UK) in accordance with the manufacturer’s instructions and data expressed as ng/mL. Serum TRACP 5b was determined using a commercially available sandwich ELISA (cat no: SB-TR103, Immunodiagnosticsystems, Tyne & Wear, UK) in accordance with the manufacturer’s instructions and data expressed as U/µL.

### 2.9. MicroCT Morphometry Analysis

Formalin-fixed tibiae from 13-week-old mice were scanned using a Skyscan 1172 X-ray microtomograph at 60 kV/167 μA with a 0.5 mm aluminium filter. Images were obtained at a 5 μm resolution with a rotation step of 0.45° at 580 ms exposure. NRecon software was used to reconstruct the images. Trabecular and cortical bone parameters were analysed using CTAn Skyscan software: Regions of interest (ROI) were selected by drawing around trabecular or cortical bone regions for each cross-sectional slice, tibia and bone architecture was determined by quantifying trabecular and cortical bone parameters using CTAn software. Trabecular bones 1.35 mm in length (200 sections) were selected for trabecular bone analysis at the metaphyseal region near the growth plate. Extent was determined by the length of trabecular bone growth in each sample, which was calculated by multiplying slice number by pixel size of scanned image (13.5 µm). Meshlab software was used to process 3D meshes of tibiae and calculate trabecular bone volume to tissue volume (BV/TV), trabecular number (Tb.N), trabecular separation (Tb.Sp) and trabecular thickness (Tb.Th).

### 2.10. Histological Analysis of Joints and Muscle

Histochemistry was performed on paraffin embedded 10 µm sections of hind paws and quadriceps of WT and TNF-tg animals following staining with haematoxylin and eosin. Pannus size at the metatarsal-phalangeal joint interface was determined using Image J software as previously reported [24]. Sections were deparaffinised and incubated in TRAP buffer for 1 h at 37 °C to detect osteoclasts. Slides were counterstained with Gills Haematoxylin in RO water (1:10) for 30 s. Quantification of osteoclasts numbers on the bone surface pannus interface of the ulna/humerus joint interface were normalised to bone surface area determined by Image J analysis of TRAP-stained paraffin-embedded sections. Sections were stained with H&E prior to quantitative analysis to visualise pannus formation at the ankle joints and CSA of fibres. For all quantifications, the mean of data from three adjacent 10 µm sections cut from the centre of the joint or from the vastus medialis from six animals were utilised and assessed using Image J software.

### 2.11. Static Histomorphometry

Static histomorphometry was performed in paraffin embedded sections. Briefly, humorous and tibia fixed in 10% neutral buffered formalin, decalcified in EDTA (0.5 M) at pH 7.4, and embedded in paraffin, and 3 μm sections were cut using a Leica Microsystems microtome (Leica Microsystems, Milton Keynes, UK). The sections were stained with either haematoxylin and eosin or tartrate-resistant acid phosphatise (TRAP) to identify osteoclasts and counterstained with Gill’s haematoxylin. The sections were examined by light microscopy (Leica Microsystems). The number of osteoblasts and osteoclasts per millimetre were measured on 6.5 mm of the corticoendosteal surfaces, starting 0.25 mm from the growth plate using the Image J analysis software (Public Domain, BSD-2).

### 2.12. Analysis of mRNA Abundance

Expression of specific mRNAs was determined using TaqMan^®^ Gene Expression Assays (Thermo Fisher Scientific, Loughborough, UK). RNA was extracted from homogenised tibia and primary cell monocultures. Briefly, for ex vivo bone, tibias were removed from the hind limb ensuring complete removal of soft tissue under a dissection microscope. The heads of bone were removed at the metaphysis and the bone marrow flushed with a syringe. The diaphysis of the tibia was powdered in liquid nitrogen in a sterilised pestle and mortar. mRNA isolation was then performed on the resulting homogenate or in primary cell monolayer using an innuPREP RNA Mini Kit (Analytikjena, Cambridge, UK) as per the manufacturer’s instructions. Aliquots containing 1 μg of RNA were then reverse transcribed using random hexamers according to the manufacturer’s protocol (4311235, Multiscribe^TM^, Thermo Fisher Scientific) to generate cDNA. The levels of murine 11β-HSD1 (*Hsd11b1*), RUNX2 (*Runx2*), OPG (*Tnfrsf11b*), RANKL (*Tnfsf11*), Osteocalcin (*Bglap*), Cathepsin K (*Ctsk*), Alkaline phosphatase (*Alp*) and Sclerostin (*Sost*) were assessed to determine expression of genes that define osteoblasts and osteoclasts and contribute to the balance of bone metabolism. Gene expression was determined using species-specific probe sets for real time quantitative PCR on the QuantStudio 5 system (Applied Biosystems, Warrington, UK). Final reactions contained 2× TaqMan PCR master mix (Life Technologies), 200 nmol TaqMan probe and 25–50 ng cDNA. The abundance of specific mRNAs in a sample was normalised to that of 18 S RNA. Data were obtained as Ct values and used to determine ΔCt values (Ct target-Ct 18 S). Data were expressed as arbitrary units using the following transformation: [arbitrary units (AU) = 1000 × (2^−Δct^)].

### 2.13. Statistical Analysis

Statistical significance was defined as *p* < 0.05 (* *p* < 0.05; ** *p* < 0.01; *** *p* < 0.001 using either an unpaired Student’s *t*-test or two-way ANOVA with a Bonferroni correction where a Gaussian distribution is identified (determined by both Kolmogorov–Smirnov and Shapiro–Wilk test), or a non-parametric Kruskal–Wallis test with a Dunn’s multiple comparison where it is absent.

## 3. Results

### 3.1. Inflammation Upregulates 11β-HSD1 in Bone and in Bone Cells

To assess the impact of a chronic inflammatory disease on 11β-HSD1 activity in bone, 11β-HSD1 activity was assessed in bone biopsies from patients with RA and osteoarthritis (OA) (Figure 1A). Patients with OA and RA had similar age, but systemic inflammation markers were higher in patients with RA (Table 1). A greater capacity to activate GCs by 11β-HSD1 was evident in RA (2-fold increase, *p* < 0.05). Primary cultures of osteoblasts and osteoclasts were generated and validated in vitro (Figure 1C,D,G,H) and 11β-HSD1 expression and activity was then assessed following their stimulation with the pro-inflammatory cytokine TNFα (Figure 1E,F,J,K). Osteoblasts cultured from OA trabecular bone were validated by positive staining for osteoid using by alizarin red and their expression of mRNA for the osteoblast markers RUNX2 and BGLAP (Figure 1B–D). Following TNFα stimulation, osteoblasts increased HSD11B1 mRNA expression and GC activation (Figure 1E,F). The combination of TNFα and GCs synergised in inducing 11β-SD1 expression and activity. In osteoclasts cultured from PBMCs, were validated based on the presence of multi-nucleated TRAP +ve cells that expressed mRNA for the mature osteoblast markers CTSK and ACP5 and their capacity to form resorption pits on dentine bone slices (Figure 1G–I). In osteoclast cultures, TNFα, increased the expression and GC activation function of 11β-HSD1 (Figure 1J,K). However, unlike in osteoblasts, the addition of activated GCs reversed the inflammatory induction of 11β-HSD1 by TNFα. Together these data reveal an upregulation of GC activation by 11β-HSD1 in bone isolated from patients with RA, and in both osteoblasts and osteoclasts in response to inflammatory stimulation with TNFα.

### 3.2. 11β-HSD1 Influences Bone Metabolism in Osteoclasts but Not Osteoblasts

To explore the contribution of 11β-HSD1 activity towards bone metabolism in primary human osteoblasts and osteoclasts, we examined its impact on their function in vitro following incubation with the inactive glucocorticoid substrate cortisone. To examine the impact of GCs activated by 11β-HSD1 on the regulation of inflammatory induced factors that promote osteoclast activation by osteoblasts, we measured their expression of RANKL and IL-6 (Figure 2A,B). TNFα potently upregulated gene expression of both factors, whilst the addition of cortisone was able to completely, and partially suppress this for RANKL (442-fold, *p* < 0.001) and IL-6 (3.3-fold, *p* < 0.05), respectively. This effect was partially rescued in response to the 11β-HSD1 inhibitor glycyrrhetinic acid (GE). Cortisone did not affect gene expression of the mature osteoblast marker BGLAP, or influence their synthesis of procollagen 1a1 at either 100 or 1000 nmol/L (Figure 2C,D). We next examined the actions of inflammatory 11β-HSD1 metabolism on osteoclast numbers and morphology. A significant decrease in osteoclast cell area and perimeter were observed in response to the addition of cortisone when administered in combination with TNFα, relative to TNFα alone, whilst total osteoclast numbers were not affected (area; TNF, 1698 ± 931 vs. TNF/E, 785 ± 186 µm, *p* < 0.05. perimeter TNF 569 ± 52 vs. TNF/E 352 ± 60, *p* < 0.05) (Figure 2E–G). The capacity for primary osteoclasts to form resorption pits in response to cortisone was assessed on bovine dentine chips. The addition of cortisone significantly reduced the global score of bone resorption by osteoclasts at 72 h (77% decrease; *p* < 0.001) (Figure 2H). Together, these findings indicate that whilst 11β-HSD1 activated GCs can influence changes in pro-osteoclastic gene expression in osteoblast cultures, it only directly influenced bone metabolism function in osteoclasts in vitro.

### 3.3. 11β-HSD1 Mediates Bone Protective Actions of Therapeutic GCs during Inflammation In Vivo

To examine the contribution of 11β-HSD1 towards protective actions of GCs in a murine model of chronic inflammation, we utilised the TNF-tg model of chronic polyarthritis receiving oral corticosterone (100 µg/mL) with global transgenic deletion of 11β-HSD1 (TNF-tg^11βKO^). Using micro-CT, we examined trabecular bone parameters in the tibia, below the epiphyseal growth plate at a site removed from local joint inflammation. TNF-tg animals receiving corticosterone were protected from inflammatory bone loss relative to vehicle treated controls with significantly higher trabecular bone volume (98.6% *p* < 0.001) and trabecular number (93.0%; *p* < 0.001), but not trabecular thickness (Figure 3A–D). Whilst a partial protection in inflammatory bone volume and trabecular number were also evident following treatment with corticosterone in TNF-tg^11βKO^ animals, this was significantly lower than in that observed in TNF-tg counterparts (BV/TV; TNF-tg/Cort, 4.25 ± 0.47 vs. TNF-tg^11βKO^/Cort, 2.82 ± 4.2, *p* < 0.01. TrN; TNF-tg/Cort, 0.00083 + 0.00004 vs. TNF-tg^11βKO^/Cort, 0.0058 ± 0.62, *p* < 0.01) (Figure 3A–D). To examine the contribution of altered bone formation by osteoblasts to this phenotype we examined procollagen type 1 N-terminal propeptide (P1NP) as a serum marker of bone formation. A trend towards decreased serum P1NP was observed in TNF-tg animals (89%; *p* = 0.071) in response to corticosterone but was absent in TNF-tg^11βKO^ counterparts (Figure 3E). In tibial mRNA isolations from these animals, the expression of Bglap (as a marker of mature osteoblasts) revealed a similar pattern, with decreased expression (6.2-fold, *p* < 0.001) following administration of corticosterone in TNF-tg animals, which was absent in TNF-tg^11βKO^ counterparts. Gene expression of the osteoblast markers Runx2 and Alpl, the wnt signalling antagonists Dkk1 and Sost, and the Rankl/Opg mRNA ratio as regulators of osteoclastogenesis did not change across groups, (Appendix A, Figure 3G). We next measured type I collagen cross-linked C-telopeptide (CTX-1) as a serum measure of osteoclast bone resorption (Figure 3G). This revealed a significant decrease in serum levels following administration of corticosterone in TNF-tg animals (56%; *p* < 0.001), but not TNF-tg^11βKO^ counterparts (Figure 3H). Whilst gene expression of the pro-osteoclast factor Csf1, and Ctsk did not change across the groups, static morphometry of TRAP+ osteoclasts in trabecular bone proximal to the epiphyseal growth plate revealed that osteoclast numbers were markedly reduced following the administration of corticosterone in TNF-tg animals (97%, *p* < 0.001) (Figure 3H,I) (Appendix A). In contrast, osteoclast numbers showed a trend towards a partial suppression in TNF-tg^11βKO^ animals receiving corticosterone (62%; NS). These data reveal a marked suppression of both osteoblastic bone formation and osteoclastic bone resorption in response to oral corticosteroids at therapeutic doses which was partially dependent on GC metabolism and activation by 11β-HSD1.

### 3.4. Animals with Mesenchymal Deletion of 11β-HSD1, Retain Bone Protective Actions of GCs

To explore the contribution of 11β-HSD1 within mesenchymal populations (including osteoblasts and osteocytes), to the corticosterone resistant bone phenotype reported in the TNF-tg^11βKO^ model, we examined TNF-tg animals with Twist2 Cre targeted deletion of 11β-HSD1 (TNF-tg^11βflx/tw2cre^). Osteoblast targeting was validated in primary calverial osteoblasts cultures derived from TNF-tg^11βflx/tw2cre^ and TNF-tg^11βKO^ animals. 11β-HSD1 activity was significantly, but not completely, attenuated in osteoblasts derived from TNF-tg^11βflx/tw2cre^ and TNF-tg^11βKO^ animals, but not in those derived from wild type controls (Figure 4A). Micro-CT analysis of the trabecular bone revealed that TNF-tg^11βflx/tw2cre^ retained a corticosteroid mediated protection from inflammatory trabecular bone loss in the tibia relative to vehicle treated controls, having greater BV/TV (51% *p* = 0.059) and trabecular number (39%; *p* = 0.061) that approached significance and closely mirrored the responses seen in TNF-tg animals (Figure 4B–E). No changes were evident in trabecular thickness (Figure 3E). A significant decrease in serum P1NP (81%. *p* < 0.05) and gene expression of the osteoblast marker Bglap (5-fold; *p* < 0.05) was apparent in TNF-tg^11βflx/tw2cre^ animals in response to corticosterone (Figure 1F,G), whilst no changes were apparent in the osteoblast markers Runx2, Alpl, Sost and Dkk1 (sup. 1G–K). Corticosterone significantly increased the Rankl/Opg mRNA ratio in TNF-tg^11βflx/tw2cre^ animals (Figure 4H). The serum marker of bone resorption, CTX-1, was also significantly decreased in TNF-tg^11βflx/tw2cre^ animals following administration of corticosterone (44% decrease; *p* < 0.05) (Figure 4I). However, whilst gene expression of the pro-osteoclast factor Csf1, and Ctsk did not change in this group in response to corticosterone, TRAP staining of trabecular osteoclasts proximal to the epiphyseal growth plate revealed that numbers were significantly reduced (93.4%; *p* < 0.01) (Figure 4J) (Appendix A). Together, these data support a bone protective effect of corticosterone in TNF-tg^11βflx/tw2cre^ animals with mesenchymal deletion of 11β-HSD1, which closely matched that observed in TNFtg counterparts.

### 3.5. Myeloid Deletion of 11β-HSD1 Causes a Partial Resistance to the Bone Protective Actions of GCs

To explore the contribution of 11β-HSD1 within myeloid derived populations (such as osteoclasts), to the corticosterone resistant bone phenotype reported in the TNF-tg^11βKO^ model, we examined TNF-tg animals with LysM Cre targeted deletion of 11β-HSD1 (TNF-tg^11βflx/LysMcre^). The efficacy of osteoclast targeted deletion of 11β-HSD1 was examined in bone marrow derived primary osteoclast cultures. This confirmed a significant reduction in 11β-HSD1 activity in osteoclasts isolated from TNF-tg^11βflx/LysMcre^ animals relative to osteoclasts isolated from wild type counterparts (8.6-fold; *p* < 0.01) (Figure 5A). Micro-CT analysis of the trabecular bone in TNF-tg^11βflx/LysMcre^ animals receiving corticosteroid revealed a marked resistance to the bone protective actions of GCs, characterised by significantly lower trabecular bone volume (31%; *p* < 0.001) and trabecular number (27%; *p* < 0.001), when compared to TNF-tg counterparts, and no increase in either parameter relative to vehicle treated controls (Figure 5B,C). No changes in trabecular thickness were apparent between groups (Figure 5C). The anti-osteoblastic properties of corticosterone administration were conserved in TNF-tg^11βflx/LysMcre^ animals with a marked decrease in serum P1NP (74.3%; *p* < 0.001) and gene expression of Bglap (7.7-fold; *p* < 0.001) (Figure 5F,G). Expression of the osteoblast markers Runx2, Alpl, Sost and Dkk1 and the ratio of the osteoclastogenic factors Rankl and Opg showed no significant changes (sup. 1G–L). Levels of CTX-1, as a measure of systemic osteoclast activity, revealed a significant decrease in TNF-tg^11βflx/LysMcre^ animals receiving corticosterone (53%; *p* < 0.01) (Figure 5I). In contrast, the suppression of osteoclast numbers by GCs proximal to the epiphyseal growth plate was markedly abrogated in TNF-tg^11βflx/LysMcre^ animals, with no significant suppression of osteoclast numbers in TNF-tg^11βflx/LysMcre^ animals receiving corticosterone relative to the vehicle treated controls (Figure 5J). Together, these data reveal that the myeloid targeted deletion of 11β-HSD1 in TNF-tg animals, may convey a partial resistance to the bone protective effects of the GC cortisone within the tibia in this inflammatory model.

## 4. Discussion

Glucocorticoids remain an important first line intervention in the management of chronic inflammatory diseases such as RA. Corticosteroids such as prednisolone are routinely utilised as a bridging therapy in newly diagnosed patients with RA, and in the management of disease flares to rapidly suppress disease activity [1,31]. We have previously shown that in a murine model of inflammatory systemic bone loss managed with oral corticosterone (to model an initial anti-inflammatory bridging therapy), GCs potently protect against inflammatory osteoclast activation and systemic inflammatory bone loss [8]. In this model, oral administration of corticosterone rapidly elevates circulating levels of the active corticosteroid corticosterone, as well as the inactive 11β-HSD1 enzyme substrate 11-DHC [15]. In this study, we explored the contribution of the GC activating enzyme 11β-HSD1 within bone to this process and examined its function and inflammatory regulation within bone using primary human tissue and cell cultures. Our study revealed a marked induction of GC activation by 11β-HSD1 within bone and bone cells in response to inflammation, with increased activity within trabecular bone in RA when compared to OA, and within osteoblasts and osteoclasts in response to the apex inflammatory cytokine TNFα. Our in vivo models revealed a central role of 11β-HSD1 in mediating the acute bone protective actions of corticosterone in animals with polyarthritis receiving therapeutic corticosteroids. In this context, the primary protective action of 11β-HSD1 appeared to be mediated by facilitating the GC induced suppression of trabecular bone loss by osteoclasts at sites distinct from the inflamed joint. The use of primary bone cell cultures and Cre targeted 11β-HSD1 knock out models supported the concept that 11β-HSD1 was mediating the intracrine activation of therapeutic GCs directly within myeloid derived osteoclasts, which in turn suppressed their capacity to resorb bone following their inflammatory activation.

The enzyme 11β-HSD1 plays a unique role during inflammatory disease, where its upregulation by mediators such as IFNγ, TNFα and IL-1β (in cell types including macrophages, stromal fibroblasts, adipocytes, muscle cells and osteoblasts) further amplify the anti-inflammatory actions of endogenous and therapeutic GCs [19,20,22,25]. This increase in 11β-HSD1 and GC activation during inflammation favours a feedback suppression of inflammatory mediators and suppression of local and systemic disease activity [26,32,33]. To date, several studies have identified 11β-HSD1 within bone cell populations, including both osteoblasts and osteoclasts in vivo [17,18]. Using primary human cultures of osteoblasts and osteoclasts and in ex vivo bone tissue, we demonstrated that this expression of 11β-HSD1 and GC activation are potently upregulated in response to pro-inflammatory mediators such as TNFα and is elevated in the bone of patients with RA. Whilst similar regulatory observations have been made in osteoblasts, the inflammatory up-regulation of 11β-HSD1 within osteoclasts is an entirely novel finding [20,33]. In the TNF-tg^11βKO^ animals, the bone protective actions of the GC corticosterone, which potently suppressed osteoclastic bone resorption, were highly dependent on 11β-HSD1. Here, global transgenic deletion of 11β-HSD1 resulted in a resistance to the bone protective effect of therapeutic corticosteroids, characterised by more severe trabecular bone loss in TNF-tg^11βKO^ animals receiving corticosterone relative to TNF-tg counterparts. This phenotype was characterised by increased serum markers of bone resorption and increased trabecular osteoclast numbers following therapeutic GC administration.

Osteoclasts play a well-defined role in mediating systemic bone loss in a chronic inflammatory disease setting, where pro-inflammatory mediators such as TNFα, IL-1β, IL-6 and RANKL promote their inflammatory activation [25,34,35]. Therapeutic interventions that suppress inflammatory bone loss can do so either indirectly, in cells such as osteoblasts, through the suppression of these pro-inflammatory mediators, or directly through their actions on the osteoclasts themselves. Therefore, we examined whether either the inflammatory upregulation of 11β-HSD1 within the osteoblast or alternatively the osteoclast mediated the marked bone protective actions of therapeutic corticosterone in our polyarthritis model.

We first examined the role of inflammatory induced 11β-HSD1 to the bone protective actions of therapeutic corticosteroids in primary cultures of osteoblasts. In this context 11β-HSD1 did not appear to directly influence anabolic osteoid formation or markers of osteoblast maturation such as RUNX2, ALP or BGLAP when exposed to the GC cortisone at therapeutic doses [26]. In contrast, 11β-HSD1 within osteoblasts, following their inflammatory activation, did influence the expression of pro-inflammatory factors such as RANKL and IL-6 that are potent activators of osteoclastic bone resorption [36]. These data suggested that 11β-HSD1 within osteoblasts might mediate the bone protective actions of therapeutic GCs indirectly in osteoclasts by suppressing production of such pro-osteoclastic factors. However, this hypothesis was not fully borne out in our in vivo models. In this study we used the TNF-tg^11βflx/tw2cre^ to selectively examine the contribution of 11β-HSD1 within mesenchymal cells and osteoblasts to our in vivo phenotypes. The Twist2 Cre targets condensed mesenchyme, with recombinase activity detected in growth plate cartilage, osteoblasts and osteocytes of the perichondrium, periosteum and endosteum, but is not present within bone marrow cells or osteoclasts [37]. This model allows for the examination of a model where 11β-HSD1 will be preserved in osteoclasts, whilst absent within myeloid populations, including within osteoblasts [15,26]. In this model, responsiveness to corticosterone was retained, with increases in BV/TV and trabecular number, and decreases in CTX-1 (as a serum marker of bone resorption) and trabecular osteoclast numbers. Furthermore, the comparable increase in the bone Rankl/Opg mRNA ratio across TNF-tg, TNF-tg^11βKO^ and TNF-tg^11βflx/tw2cre^ animals did not appear to be predictive of an increase in indirect mediators of bone resorption by osteoclasts in the Twist2 Cre. Collectively, these data suggested that 11β-HSD1 in osteoblasts did not appear to be a predominant indirect mediator of the bone protective anti-osteoclastic actions of therapeutic GCs through the modulation of pro-inflammatory osteoclast activating factors such as RANKL and IL-6.

Therefore, we next examined whether 11β-HSD1 may be mediating the bone protective effects of corticosteroids through its expression and inflammatory upregulation within osteoclasts. Unlike in osteoblasts, in vitro assessment in primary human osteoclasts revealed that GC activation by 11β-HSD1 had a direct effect in mediating reduced osteoclastogenesis and bone resorption. To examine this in more detail, we utilised the myeloid targeted LysM Cre to knock out 11β-HSD1 within myeloid derived osteoclasts in our TNF-tg polyarthritis model treated with corticosteroids. The LysM-Cre system is one of the most commonly used models to target the osteoclastic lineage, but also has the caveat that it targets alternative myeloid derived cell populations including macrophages and neutrophils [38].

Our in vitro analysis confirmed significant suppression of GC activation by 11β-HSD1 in bone marrow osteoclasts derived from TNF-tg^11βflx/LysMcre^ animals. These animals presented with a partial abrogation of the bone protective actions of therapeutic GCs in vivo. This was underpinned by an increase in osteoclast numbers in trabecular bone proximal to the epiphyseal growth plate in TNF-tg^11βflx/LysMcre^ animals receiving cortisone relative to TNF-tg, where corticosterone robustly suppresses osteoclast numbers and prevents inflammatory bone loss. These data indicate that in both the TNF-tg^11βKO^ and TNF-tg^11βflx/LysMcre^ models, the loss of 11β-HSD1, and its capacity to convert inactive 11-DHC to the active corticosterone, limits the bone protective effects of oral corticosterone. These findings reveal a degree of GC resistance in these TNF-tg^11βflx/LysMcre^ osteoclasts and support the concept that 11β-HSD1 mediates the bone protective actions of corticosteroids during inflammation directly within osteoclasts. Whilst these findings could be further validated in vitro, to directly delineate the role of 11β-HSD1 on bone resorption in bone marrow derived osteoclasts from the TNF-tg^11βflx/LysMcre^ animals, this went beyond the scope of the current study.

It is important to recognise the limitations of these transgenic and Cre targeted 11β-HSD1 knock out models. Both the TNF-tg^11βflx/tw2cre^ and TNF-tg^11βflx/LysMcre^ models have off target deletion outside of the osteoblast and osteoclast lineage that may indirectly influence the protective corticosteroid bone phenotype. Furthermore, we have previously reported that TNF-tg^11βKO^ animals have increased disease severity that could contribute to increased systemic inflammatory bone loss as a confounder [26,39]. To compensate for this latter point, we harvested bones at an earlier time point in this disease model, at 7 weeks, when a significant difference in disease severity was not apparent between models. Full disease severity and inflammation scores and histology for these animals at this stage are reported by Fenton et al. [15]. Whilst future studies might focus on alternative Cre targeting systems, such as the Col1a1 Cre for osteoblasts or the CtsK-Cre for osteoclasts, this current dataset in combination with the in vitro functional data, strongly suggest that 11β-HSD1 within osteoclasts mediates the bone protective actions of therapeutic GCs in vivo.

Whilst therapeutic GCs have proven beneficial in the initial suppression of inflammatory bone loss associated with chronic inflammation, it is important to note that, GCs also suppress osteoid deposition and bone formation in osteoblasts through the induction of apoptosis and autophagy within osteoblasts [8,40,41,42,43]. The contribution of 11β-HSD1 to the anti-anabolic actions of GCs on bone have also been examined in models of therapeutic corticosteroids administration in this setting in the absence of inflammation. Here, its transgenic deletion or therapeutic inhibition prevent GIOP by preserving osteoblast numbers and anabolic bone formation [8,9,44]. These studies reveal that under conditions of corticosteroid excess, inhibition of 11β-HSD1 protects bone from the damaging effects of GCs on anabolic bone formation by osteoblasts. In our inflammatory models, the benefits of 11β-HSD1 inhibition on GIOP were less apparent, where the predominant actions of corticosteroids were bone protective through there suppression of inflammatory bone resorption. However, of interest, our models did reveal that 11β-HSD1 still mediated these anti-anabolic actions of GCs in osteoblasts. In this setting, its global deletion in TNF-tg^11βKO^ animals abrogated the suppression of bone formation markers such as P1NP and Bglap. Consequently, this is the first study to demonstrate a comparable anabolic effect of 11β-HSD1 inhibition in a model of systemic inflammation. However, these effects were paradoxical to the parallel loss of the bone protective actions of therapeutic GCs that favoured inflammatory bone loss in our model.

## 5. Summary

These data reveal novel insights into the regulation and role of the GC activating enzyme 11β-HSD1 within bone and bone cells in a chronic inflammatory disease. Namely, the pro-inflammatory induction of 11β-HSD1 within osteoclasts and their myeloid precursors, increases their local reactivation and amplification of therapeutic corticosteroids to rapidly shut down inflammatory bone loss and mediate the bone protective actions of therapeutic GCs in a chronic inflammatory disease setting.

## Figures and Tables

**Figure 1 ijms-23-07334-f001:**
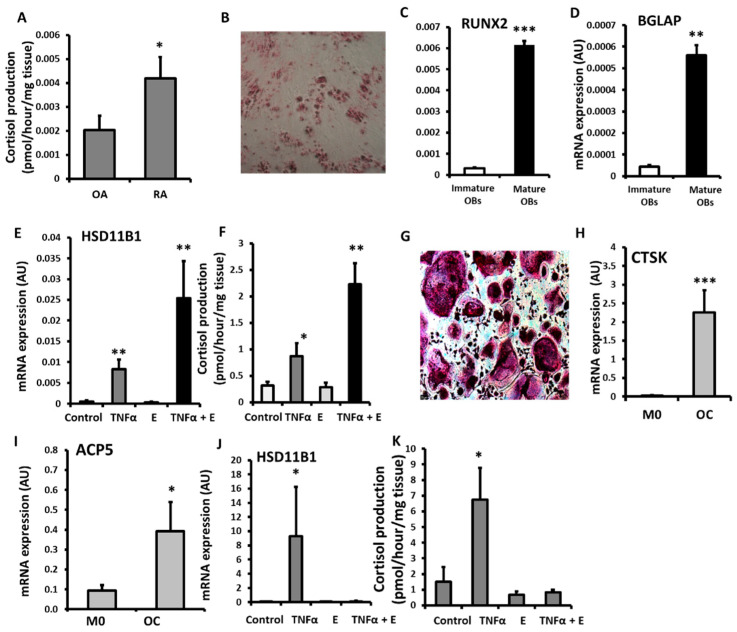
(**A**) Glucocorticoid activation by 11βHSD1 in ex vivo tibia explants freshly isolated after joint replacement surgery from patients with rheumatoid arthritis (RA, *n* = 8) and osteoarthritis (OA *n* = 9) determined by scanning thin-layer chromatography. (**B**) Alizarin red staining of osteoid nodules in primary human osteoblast (OBs) culture isolated from OA tibia. (**C**) RUNX2, (**D**) BGLAP and (**E**) HSD11B1 mRNA expression (AU) determined by RT qPCR and (**F**) glucocorticoid activation by 11βHSD1 in primary human osteoblast cultures isolated from OA tibia following treatments with either TNFα (10 ng/mL), cortisone (denoted E) (1000 nmol/L) or a combination of both for 24 h. (**G**), TRAP staining of multinucleated primary human osteoclast (OC) culture isolated from peripheral blood mononuclear cells. (**H**) CTSK and (**I**) ACP5 mRNA expression (AU) determined by RT qPCR in primary human macrophages (M0) and osteoclasts (OC) cultures isolated from peripheral blood mononuclear cells. (**J**) HSD11B1 expression (AU) determined by RT qPCR, and (**K**) glucocorticoid activation by 11βHSD1 in primary human osteoclast culture isolated from peripheral blood mononuclear cells following treatments with either TNFa (10 ng/mL), cortisone (1000 nmol/L) or a combination of both for 24 h. Values are expressed as mean ± standard error, *n* = 4 per group for primary culture. Statistical significance was determined using either two-way or one-way ANOVA with Tukey post hoc analysis. * *p* <  0.05, ** *p* <  0.01 and *** *p* <  0.001.

**Figure 2 ijms-23-07334-f002:**
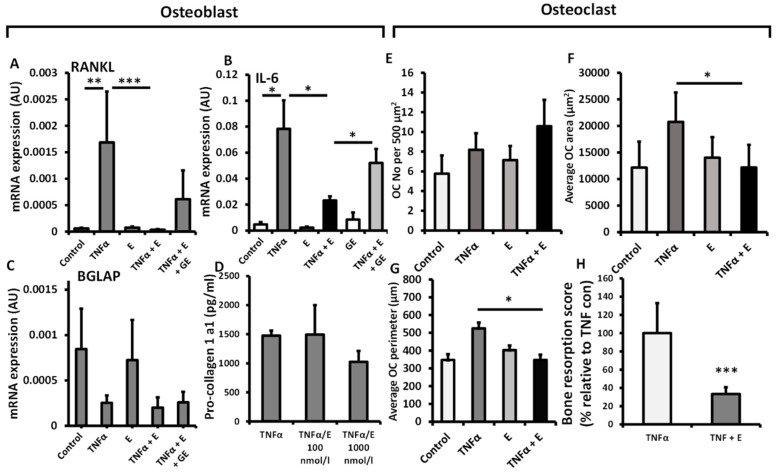
mRNA **e**xpression (AU) of (**A**) TNFSF11, (**B**) IL6 and (**C**) BGLAP determined by RT qPCR in primary human osteoblast culture isolated from OA tibia, following treatment with either TNFα (10 ng/mL), cortisone (denoted E) (1000 nmol/L), glycyrrhetinic acid (GE) (1 µM) or a combination of the above for 24 h. (**D**) Pro-collagen 1 synthesis by primary human osteoblast cultures isolated from OA tibia following treatment with either TNFα (10 ng/mL), cortisone (100 nmol/L), glycyrrhetinic acid (1 µM) or a combination of the above for 48 h. (**E**) Numbers, (**F**) average area and (**G**) average perimeter of primary human osteoclast cultures isolated from peripheral blood mononuclear cells determined in vitro by light microscopy and using Image J analysis per 500 µm^2^, following treatment with TNFα (10 ng/mL), cortisone (1000 nmol/L) or a combination of both for 72 h. (**H**) In vitro scoring of bone erosions on dentine chips following incubation with primary human osteoclasts (isolated from peripheral blood mononuclear cells) after 48 h determined by histological scoring following treatment with either TNFα (10 ng/mL), or a combination of TNFα and cortisone (100 nmol/L). Values are expressed as mean ± standard error, *n* = 4 per group for primary culture. Statistical significance was determined using either two-way or one-way ANOVA with Tukey post hoc analysis. * *p* <  0.05, ** *p* <  0.01 and *** *p* <  0.001.

**Figure 3 ijms-23-07334-f003:**
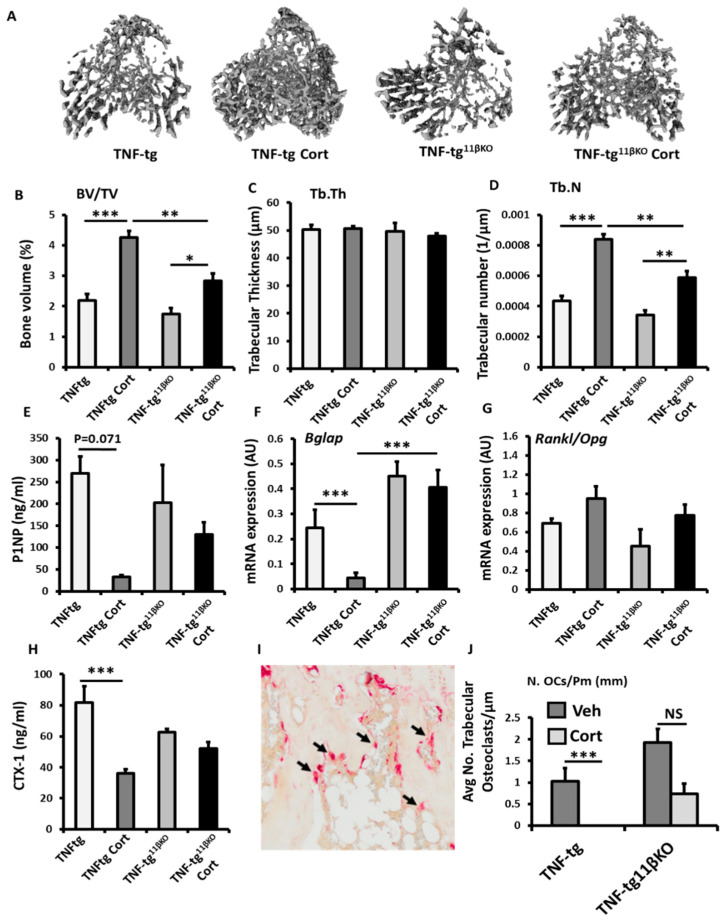
(**A**) Representative images of 3D reconstructions of tibia trabecular bone using micro-CT, (**B**) bone volume to tissue volume (BV/TV), (**C**) trabecular thickness (Tb.Th) and (**D**) trabecular number (Tb.N) determined by Meshlab software analysis of micro CT in TNF-tg and TNF-tg^11βKO^ animals receiving either vehicle (Veh) or corticosterone (Cort) (100 µg/mL) in drinking water over 3 weeks. (**E**) Serum P1NP (ng/mL), mRNA expression (AU) of (**F**) Bglap mRNA expression (AU), (**G**) RankL/Opg mRNA ratio in trabecular bone and (**H**) serum CTX-1 (ng/mL) in TNF-tg and TNF-tg^11βKO^ animals receiving either vehicle or corticosterone (100 µg/mL) in drinking water over 3 weeks and determined by ELISA and RT-qPCR. (**I**) Representative TRAP staining of trabecular osteoclasts on the corticoendosteal surfaces proximal of the epiphyseal growth plate in formalin fixed 5 µm sections, and (**J**) average number of osteoclasts normalised to trabecular bone perimeter (N. OCs/Pm (mm)) in TNF-tg and TNF-tg^11βKO^ animals receiving either vehicle or corticosterone (100 µg/mL) in drinking water over 3 weeks determined using Image J analysis software following TRAP staining. Values are expressed as mean ± standard error of six animals per group. Statistical significance was determined using two-way ANOVA with a Tukey post hoc analysis. * *p* < 0.05, ** *p* < 0.005 and *** *p* < 0.001.

**Figure 4 ijms-23-07334-f004:**
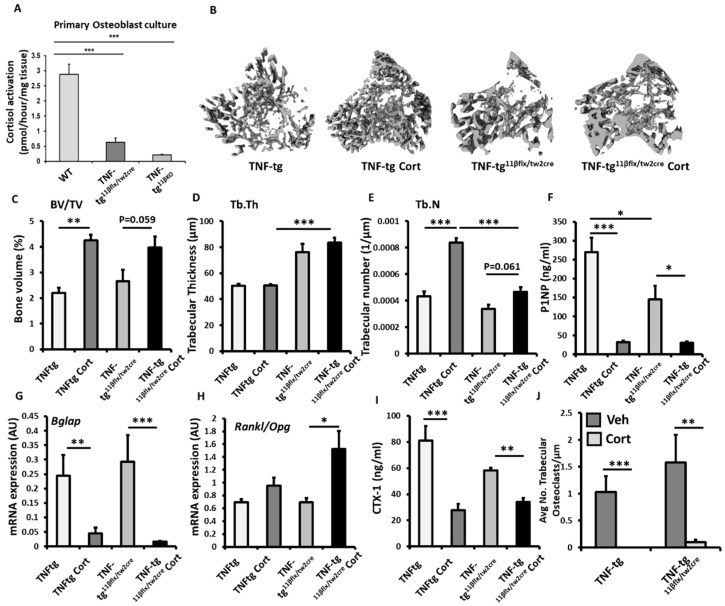
(**A**) Glucocorticoid activation by 11β-HSD1 over 24 h in primary murine calvaria osteoblast cultures isolated from wild type (WT), TNF-tg and TNF-tg^11βflx/tw2cre^animals determined by scanning thin layer chromatography. (**B**) Representative images of 3D reconstructions of tibia trabecular bone using micro-CT, (**C**) bone volume to tissue volume (BV/TV), (**D**) trabecular thickness (Tb.Th) and (**E**) trabecular number (Tb.N) determined by Meshlab software analysis of micro CT in TNF-tg and TNF-tg^11βflx/tw2cre^ animals receiving either vehicle (Veh) or corticosterone (Cort) (100 µg/mL) in drinking water over 3 weeks. (**F**) Serum P1NP (ng/mL), mRNA expression (AU) of (**G**) Bglap mRNA expression (AU), (**H**) RankL/Opg mRNA ratio in trabecular bone and (**I**) serum CTX-1 (ng/mL) in TNF-tg and TNF-tg^11βflx/tw2cre^ animals receiving either vehicle or corticosterone (100 µg/mL) in drinking water over 3 weeks determined by RT-qPCR or ELISA. (**J**) Average number of osteoclasts normalised to trabecular bone perimeter (N.Oc/Pm (mm)) in TNF-tg and TNF-tg^11βflx/tw2cre^ animals receiving either vehicle or corticosterone (100 µg/mL) in drinking water over 3 weeks determined using Image J analysis software following TRAP staining. Values are expressed as mean ± standard error of six animals, or 3 primary cell cultures per group. Statistical significance was determined using two-way ANOVA with a Tukey post hoc analysis. * *p* < 0.05, ** *p* < 0.005 and *** *p* < 0.001.

**Figure 5 ijms-23-07334-f005:**
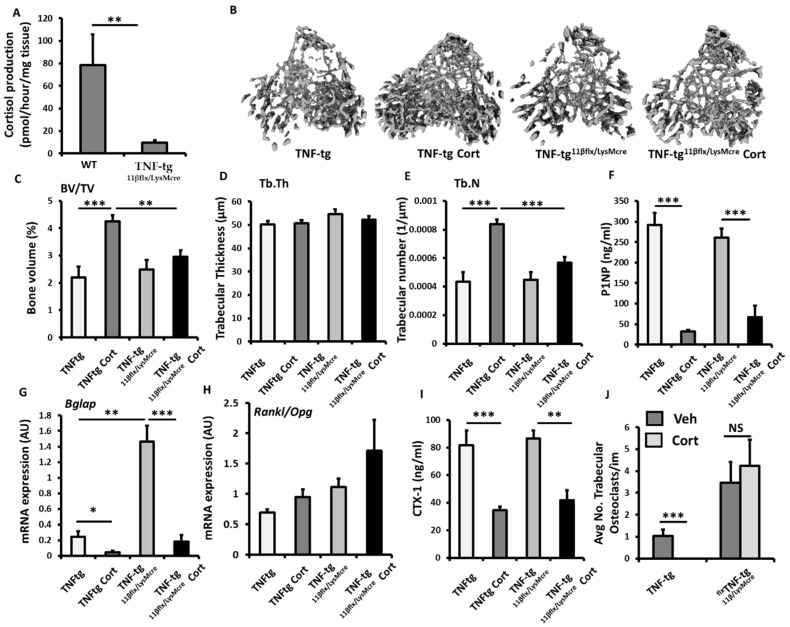
(**A**) Glucocorticoid activation by 11β-HSD1 over 24 h in primary bone marrow derived osteoclast cultures isolated from wild type, and TNF-tg^11βflx/LysMcre^ animals determined by scanning thin layer chromatography. (**B**) Representative images of 3D reconstructions of tibia trabecular bone using micro-CT, (**C**) bone volume to tissue volume (BV/TV), (**D**) trabecular thickness (Tb.Th) and (**E**) trabecular number (Tb.N) determined by Meshlab software analysis of micro CT in TNF-tgand TNF-tg^11βflx/LysMcre^ animals receiving either vehicle (Veh) or corticosterone (Cort) (100 µg/mL) in drinking water over 3 weeks. (**F**) Serum P1NP (ng/mL), mRNA expression (AU) of (**G**) Bglap mRNA expression (AU), (**H**) RankL/Opg mRNA ratio and (**I**) serum CTX-1 (ng/mL) in TNF-tg and TNF-tg^11βflx/LysMcre^ animals receiving either vehicle or corticosterone (100 µg/mL) in drinking water over 3 weeks determined by either RT-qPCR or ELISA. (**J**) Average number of osteoclasts normalised to trabecular bone perimeter (N.Oc/Pm (mm) in TNF-tg and TNF-tg^11βflx/LysMcre^ animals receiving either vehicle or corticosterone (100 µg/mL) in drinking water over 3 weeks determined using Image J analysis software following TRAP staining. Values are expressed as mean ± standard error of six animals, or 3 primary cell cultures per group. Statistical significance was determined using two-way ANOVA with a Tukey post hoc analysis. * *p* < 0.05, ** *p* < 0.005 and *** *p* < 0.001.

**Table 1 ijms-23-07334-t001:** Patients’ characteristics for human skeletal muscle biopsies. Biopsies were collected from patients with either rheumatoid arthritis (RA) or osteoarthritis (OA) following elective joint replacement surgery. Patient age, C-reactive protein (CRP), erythrocyte sedimentation rate (ESR) and concurrent therapeutic interventions were recorded prior to surgery.

Patient Details	Patients with OA (*n* = 12)	Patients with RA (*n* = 10)	Group Comparison (*p*-Value)
Age (years)	66.2 + 3.3	65.7 + 4.1	0.93
CRP (mg/L)	2.3 + 1.7	11.0 + 3.5	0.03
ESR (mm/h)	1.9 + 1.0	24.13 + 7.1	0.001
Methotrexate (*n*)	0	5	na
Anti-TNFα therapy (*n*)	0	2	na
Prednisolone	0	0	na

## Data Availability

Data and materials within this manuscript can be made available upon reasonable request to the corresponding author.

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
