# Peer review of "11β-Hydroxysteroid Dehydrogenase Type 1 within Osteoclasts Mediates the Bone Protective Properties of Therapeutic Corticosteroids in Chronic Inflammation"

_ijms, 2022, doi:10.3390/ijms23137334_

Round 1

Reviewer 1 Report

Authors proposed critical role of 11β-HSD1 in mediating the suppression of inflammatory bone resorption at chronic inflammatory situation. However, your manuscript could be developed a bit further. 

Comments.

Major.

1.     First of all, manuscript and figure legends are poorly organized and are very difficult to decipher. In particular, the figure legend should be described more systematically. For example, abbreviations are not described precisely. What does “OBs” mean? What does “Con”, “E”, “M0” and “OC” mean in figure1?  What do “E”, “GE”, “E1000”, “Cort”, “AVG OSTC”, “TNF11βKO”, “VEH”, etc. mean? "TNF E", "TNF Cort", "TNF-LysM Cort", "TNF-Twist2 Cort", etc. are very difficult to understand what is meant by these descriptions. The experimental procedure should also be more obvious in the figure legend.  In other words, it is necessary to describe whether it is primary culture, measurement of serum, or analysis of extracts from tibia. These pieces of information are missing in some places.

2.     L42, Authors concluded “These data reveal the critical role of 11β-HSD1 within bone and osteoclasts in mediating the suppression of inflammatory bone resorption in response to therapeutic GCs in chronic inflammatory disease”. However, I cannot find the data of bone resorption. Although 11β-HSD1 effect the number of osteoclasts, bone resorption markers are similarly suppressed by steroids in both conditional knockout mice and wild type (fig 5I). You need bone resorption assay using osteoclasts from TNF-LysM mouse.

3.     Although I agree that 11βHSD1 in osteoclasts is important in the phenomenon of corticosterone-induced osteoclast reduction in the chronic inflammatory situation, there are some aspects which I disagree about that mechanism. In the manuscript, authors mainly claim that this phenomenon is the result of the enzymatic activity of 11βHSD1.  However, Fig. 5J shows that corticosterone-induced osteoclast reduction does not occur when 11βHSD1 is deleted in osteoclasts. Because corticosterone is the metabolites by 11βHSD1, Fig. 5J indicates that the enzymatic activity of 11βHSD1 that activates GC precursors is not involved in this phenomenon.

Minor

1.     In line 260, authors described “their 11β-HSD1 expression and activity assessed following stimulation with the pro-inflammatory cytokine TNFα (fig.1b-k).” However, TNFα is not associated with 1c, 1d, and 1g, 1h.

2.     In line 274, does “Figure 1. Glucocorticoid activation by 11βHSD1 in ex vivo tibia explants” mean “Figure 1. (A) Glucocorticoid activation…………….”?

3.     In line 279, I cannot understand the description “(G), TRAP staining of multinucleated primary human osteoclast culture isolated from peripheral blood mononuclear cells. Rred staining of osteoid nodules in primary human osteoblast culture isolated from OA tibia.” Is this osteoclast culture, or osteoblast culture?

4.     In figure 2E, bars on “Con” and “TNF” consist of two patterns. What do these patterns mean?

Author Response

We would like to thank the reviewers for their comments and insights towards this manuscript. All recommended changes have been addressed as follows:

Reviewer 1

Authors proposed critical role of 11β-HSD1 in mediating the suppression of inflammatory bone resorption at chronic inflammatory situation. However, your manuscript could be developed a bit further. 

Comments.

Major.

  1. First of all, manuscript and figure legends are poorly organized and are very difficult to decipher. In particular, the figure legend should be described more systematically. For example, abbreviations are not described precisely. What does “OBs” mean? What does “Con”, “E”, “M0” and “OC” mean in figure1?  What do “E”, “GE”, “E1000”, “Cort”, “AVG OSTC”, “TNF11βKO”, “VEH”, etc. mean? "TNF E", "TNF Cort", "TNF-LysM Cort", "TNF-Twist2 Cort", etc. are very difficult to understand what is meant by these descriptions. The experimental procedure should also be more obvious in the figure legend.  In other words, it is necessary to describe whether it is primary culture, measurement of serum, or analysis of extracts from tibia. These pieces of information are missing in some places.

Thank you for highlighting this important point in relation to clarity of the manuscript. We have now addressed the description of the abbreviations and clarified the information in the figure legends. All changes to manuscript legends and text are highlighted in red throughout.

  1. L42, Authors concluded “These data reveal the critical role of 11β-HSD1 within bone and osteoclasts in mediating the suppression of inflammatory bone resorption in response to therapeutic GCs in chronic inflammatory disease”. However, I cannot find the data of bone resorption. Although 11β-HSD1 effect the number of osteoclasts, bone resorption markers are similarly suppressed by steroids in both conditional knockout mice and wild type (fig 5I). You need bone resorption assay using osteoclasts from TNF-LysM mouse.

Thank you for you raising this point. These statements were based on previous studies that have demonstrated that elevated osteoclast numbers closely reflect increased systemic bone resorption and systemic bone loss in patients with chronic inflammatory diseases such as rheumatoid arthritis (RA) (Green M J, PMID: 11358418) and in these murine models of polyarthritis (Fenton CM, PMID: 31370858). Consequently, we did link the observation that the changes in osteoclast numbers were impacting on bone resorption, given that 11β-HSD1 knock out models abrogated the bone protective effects of oral corticosteroids by micro-CT. However, we recognize that from our current in vivo dataset we cannot conclusively validate this without in vitro osteoclast resorption assays isolated from the TNF-LysM mouse. Unfortunately, we no longer have this animal cre line running within our facilities and the time and costs required for re-derivatization of this triple transgenic animal now go beyond the scope of what can be achieved in this study. However, to accurately reflect your point we have reworded the central conclusions to more accurately reflect the actual experimental result and highlight this further important experiment as follows:

 Abstract Line 42 “These data reveal the critical role of 11β-HSD1 within bone and osteoclasts in mediating the suppression of inflammatory bone loss in response to therapeutic GCs in chronic inflammatory disease.”

 Discussion: Line 551 “Whilst these findings could be further validated in vitro, to directly delineate the role of 11β-HSD1 on bone resorption in bone marrow derived osteoclasts from the TNF-tg11βflx/LysMcre animals, this went beyond the scope of the current study.”

 Discussion: Line 591 “These data reveal novel insights into the regulation and role of the GC activating en-zyme 11β-HSD1 within bone and bone cells in a chronic inflammatory disease. Namely, the pro-inflammatory induction of 11β-HSD1 within osteoclasts and their myeloid pre-cursors, increases their local reactivation and amplification of therapeutic corticosteroids to rapidly shut down inflammatory bone loss and mediate the bone protective actions of therapeutic GCs in a chronic inflammatory disease setting.”

  1. Although I agree that 11βHSD1 in osteoclasts is important in the phenomenon of corticosterone-induced osteoclast reduction in the chronic inflammatory situation, there are some aspects which I disagree about that mechanism. In the manuscript, authors mainly claim that this phenomenon is the result of the enzymatic activity of 11βHSD1.  However, Fig. 5J shows that corticosterone-induced osteoclast reduction does not occur when 11βHSD1 is deleted in osteoclasts. Because corticosterone is the metabolites by 11βHSD1, Fig. 5J indicates that the enzymatic activity of 11βHSD1 that activates GC precursors is not involved in this phenomenon.

Apologies for any confusion here. When administered in vivo, corticosterone (secondary to renal and hepatic metabolism) raises both the serum corticosterone (active) and 11-dehydrocorticosterone (11-DHC) (inactive 11BHSD1 substrate) concentrations within the circulation (Fenton ann rheum dis 2020). In figure 5J, we show that in animals with intact 11BHSD1 (TNFtg animals highlighted in red below) administration of corticosterone (which raises 11DHC serum levels) suppresses osteoclast numbers in bone. In both the global and LysMcre 11BHSD1 knock out animals (highlighted in blue below), this suppression of osteoclast numbers is lost, demonstrating the importance of 11BHSD1 in mediating this effect. Given that 11-DHC is an inactive steroid, this effect could only be mediated by the reactivation and amplification of corticosterone from 11-DHC by 11BHSD1. We have clarified this by making clearer indications of the comparisons made in the figures and further interpretation within the discussion as follows:

 Discussion Line 461” In this model, oral administration of corticosterone rapidly elevates circulating levels of the active corticosteroid corticosterone, as well as the inactive 11β-HSD1 enzyme substrate 11-dehydrocortciosterone (15).”

 Discussion Line 546 “This was underpinned by an increase in osteoclast numbers in trabecular bone proximal to the epiphyseal growth plate in TNF-tg11βflx/LysMcre animals receiving cortisone relative to TNF-tg, where corticosterone robustly suppresses osteoclast numbers and prevents inflammatory bone loss. These data indicate that in both the TNFtg11βKO and TNF-tg11βflx/LysMcre models, the loss of 11β-HSD1 and its capacity to convert inactive 11-DHC to the active corticosterone, limit the bone protective effects of oral corticosterone.”

Minor

  1. In line 260, authors described “their 11β-HSD1 expression and activity assessed following stimulation with the pro-inflammatory cytokine TNFα (fig.1b-k).” However, TNFα is not associated with 1c, 1d, and 1g, 1h.

The reference to the figure was updated to match the figure and the changes are highlighted in red in the manuscript.

  1. In line 274, does “Figure 1. Glucocorticoid activation by 11βHSD1 in ex vivo tibia explants” mean “Figure 1. (A) Glucocorticoid activation…………….”?

 Thank you for highlighting this important point in line 274.The reference to the figure was updated to match the figure and the changes are highlighted in red in the manuscript as follows.

 “Primary cultures of osteoblasts and osteoclasts were generated and validated in vitro (fig.1c,d,g,h) and 11β-HSD1 expression and activity was then assessed following their stimulation with the pro-inflammatory cytokine TNFα (fig.1e,f,j,k).”

  1. In line 279, I cannot understand the description “(G), TRAP staining of multinucleated primary human osteoclast culture isolated from peripheral blood mononuclear cells. Rred staining of osteoid nodules in primary human osteoblast culture isolated from OA tibia.” Is this osteoclast culture, or osteoblast culture?

Apologies. This sentence is a duplication and has been deleted from the line 279 as follows:

“(G), TRAP staining of multinucleated primary human osteoclast culture isolated from peripheral blood mononuclear cells.  Red staining of osteoid nodules in primary human osteoblast culture isolated from OA tibia.”

  1. In figure 2E, bars on “Con” and “TNF” consist of two patterns. What do these patterns mean?

The graphs were generated in a different analytical program and the patterns have no specific meaning. To address the difference in style, graphs were redone to match the rest of the manuscript

Reviewer 2 Report

This study is presenting interesting results in the field of glucocorticoid effect on bone health. However, some aspects of the manuscript should be amended:

- Methods section is poorley organized - prober numeration of relevant sub-sections should be incorporated

- p-values should be always written with 3 decimal places

- in Table 1, legend and abbrevations are missing

- Results should be reorganized as textual part is too large - it should be written in more concise manner, and more directed to relevant findings of this study, without citing other literature and explaining earlier findings

Author Response

We would like to thank the reviewers for their comments and insights towards this manuscript. All recommended changes have been addressed as follows:

Reviewer 2

This study is presenting interesting results in the field of glucocorticoid effect on bone health. However, some aspects of the manuscript should be amended:

- Methods section is poorley organized - prober numeration of relevant sub-sections should be incorporated

Apologies for this formatting error. This has been addressed throughout

- p-values should be always written with 3 decimal places

This has been addressed where whole p values are quoted

 - in Table 1, legend and abbrevations are missing

These have been updated

- Results should be reorganized as textual part is too large - it should be written in more concise manner, and more directed to relevant findings of this study, without citing other literature and explaining earlier findings

Recognising this important point, we have undertaking significant editing of the results chapters to reduce non-essential text, remove citations and references to previous findings and ensure the essential data to the project are discussed in a concise fashion. For clarity of reading, non-essential text was deleted with further edits or inclusions highlighted in red. 

Round 2

Reviewer 1 Report

Thank you for your reply. Regarding my comments 1 and 2, I agree with the authors' explanations.

I do not agree with your reply about comment 3. In this setting, corticosterone (active) is present in the body. I cannot understand why this corticosterone (active) did not suppress osteoclast numbers in bone.

The authors assume that only intracellularly activated corticosterone has a function in this system? In other words, why extracellular corticosterone does not function, and only intracellularly activated corticosterone function in this setting? This phenomenon is not self-evident, thus authors need data or explanation about this point.

One of the other possibilities is concentration of corticosterone (active). In this case also, you need data or explanation about this point.

It is possible that the nature of osteoclast has changed so that it does not respond to corticosterone (active).

I think the argument needs to include these possibilities.

Author Response

1:         I do not agree with your reply about comment 3. In this setting, corticosterone (active) is present in the body. I cannot understand why this corticosterone (active) did not suppress osteoclast numbers in bone. The authors assume that only intracellularly activated corticosterone has a function in this system? In other words, why extracellular corticosterone does not function, and only intracellularly activated corticosterone function in this setting? This phenomenon is not self-evident, thus authors need data or explanation about this point. One of the other possibilities is concentration of corticosterone (active). In this case also, you need data or explanation about this point. It is possible that the nature of osteoclast has changed so that it does not respond to corticosterone (active).I think the argument needs to include these possibilities.

We agree that further clarification of the possibilities will help with the interpretation of the manuscript. In particular, we believe (as you highlight) that the relative concentrations of circulating corticosterone are central to this phenomenon. This is evidenced from observations in our seminal study examining the anti-inflammatory properties of corticosterone (active) in murine models of arthritis. (Fenton, 2020, ann rheum dis. PMID: 33162397)

Here, we revealed that whilst some effects of corticosterone were independent of 11BHSD1 and mediated purely by circulating levels of corticosterone (ie reduced synovial infiltration of CD8+ and CD19+ cells), other actions were dependant on 11BHSD1 (infiltration of neutrophils, macrophages) (fenton, 2020, ann rheum dis, figure 3 appended below).

These 11BHSD1 dependent effects also happened in the presence of measurable levels of circulating corticosterone. However, when the dose of oral corticosterone was titrated down, the effects of circulating corticosterone were lost, demonstrating that their actions were dose dependent in these cells. These observation are supported by studies showing that the anti-inflammatory properties of GCs are dose dependent in leukocytes like macrophages  (Dong et al PMID: 29378573).

Furthermore, when we isolated the corticosterone resistant primary macrophages from 11BHSD1 KO animals we were able to demonstrate that they retain their responsiveness to active corticosterone (when the GC responsive gene gilz was assessed) demonstrating that they are still GC responsive (supp fig attached below). In contrast, they were resistant to inactive 11dehydrocorticosterone. Consequently, the conclusion from the Fenton 2020 study was that 11BHSD1 facilitated the amplification of corticosterone action rather than the cells were resistant to active corticosterone.

However, in the absence of comparable in vitro experiments in osteoclasts from these animals we have also included a line stating that a shift in osteoclast properties favoring a resistance to the active glucocorticoid may have occurred in the 11BHSD1 KO animal. Within the manuscript, the following paragraph has been amended to address the above points:

Line 553: “Our in vitro analysis confirmed significant suppression of GC activation by 11β-HSD1 in bone marrow osteoclasts derived from TNF-tg11βflx/LysMcre animals. These animals presented with a partial abrogation of the bone protective actions of therapeutic GCs in vivo. This was underpinned by an increase in osteoclast numbers in trabecular bone proximal to the epiphyseal growth plate in TNF-tg11βflx/LysMcre animals receiving cortisone relative to TNF-tg, where corticosterone robustly suppresses osteoclast numbers and prevented inflammatory bone loss. These data indicate that in both the TNFtg11βKO and TNF-tg11βflx/LysMcre models, the loss of 11β-HSD1, and its capacity to convert inactive 11-DHC to the active corticosterone, limit the bone protective effects of oral corticosterone. This GC resistant phenotype mirrored observations within alternative cell populations, such as macrophages and neutrophils in the TNFtg mouse, where transgenic deletion of 11β-HSD1 conveys resistance to the anti-inflammatory actions of oral corticosterone (15). Of interest, this GC resistant phenotype occurred despite detectable levels of circulating active corticosterone. In the study by Fenton et al, the reason for this was dependent on the circulating corticosterone dose, which could be titrated out with lower oral delivery, and matched dose dependent effects of GCs in these cells in vitro (15, 49). Therefore, these studies suggested that the autocrine cellular amplification of corticosterone from 11-DHC is likely a central driver of the oral corticosterone effect in these cell populations. Consequently, the findings of this study may suggest that a degree of GC resistance in the TNF-tg11βflx/LysMcre osteoclasts may occur secondary to the loss of this 11β-HSD1 amplification step within these cells. Therefore, these findings would support the concept that 11β-HSD1 mediates the bone protective actions of corticosteroids within osteoclasts in this context. An alternative explanation might be that the deletion of 11β-HSD1 alters the maturation or functional properties of these osteoclasts in vivo, resulting in their becoming resistant to the actions of circulating corticosterone. Whilst these different possibilities could be further validated using cultures derived from TNF-tg11βflx/LysMcre animals, these experiments went beyond the scope of the current study.

Round 3

Reviewer 1 Report

Thank you for your response.

In this revision, this manuscript describes the authors' arguments in a logical and convincing manner.